# Effects of Regulating Visitor Viewing Proximity and the Intensity of Visitor Behaviour on Little Penguin (*Eudyptula minor*) Behaviour and Welfare

**DOI:** 10.3390/ani9060285

**Published:** 2019-05-28

**Authors:** Samantha J. Chiew, Kym L. Butler, Sally L. Sherwen, Grahame J. Coleman, Kerry V. Fanson, Paul H. Hemsworth

**Affiliations:** 1Animal Welfare Science Centre, Faculty of Veterinary and Agricultural Sciences, University of Melbourne, North Melbourne, Victoria 3051, Australia; kym.butler@unimelb.edu.au (K.L.B.); grahame.coleman@unimelb.edu.au (G.J.C.); phh@unimelb.edu.au (P.H.H.); 2Biometrics Team, Agriculture Victoria Research, Department of Jobs, Precincts and Regions, Hamilton, Victoria 3300, Australia; 3Department of Wildlife Conservation and Science, Zoos Victoria, Parkville, Victoria 3052, Australia; ssherwen@zoo.org.au; 4Centre for Integrative Ecology, Deakin University, Geelong, Victoria 3216, Australia; kerryfanson@gmail.com

**Keywords:** little penguins, visitor–animal interactions, zoos, visitor effect

## Abstract

**Simple Summary:**

Some studies on zoo penguins have shown that the presence of visitors may be fear-provoking. To manage zoo visitor–animal interactions, it is important to determine what it is about visitors that affects zoo penguins. Consequently, we investigated the effects of regulating both zoo visitor viewing proximity to the penguin enclosure and the intensity of visitor behaviour on the behaviour and stress physiology of little penguins. A physical barrier was used to increase visitor viewing distance, by 2 m away from the enclosure, and signage and uniformed personnel was used to attempt to regulate the intensity of visitor behaviour. Signage and uniformed personnel had no effect on visitor number, noise level or visitor behaviour. However, the physical barrier reduced the frequency of potentially threatening visitor behaviours such as banging on enclosure features, looming over the pool, physical contact with the pool’s water and sudden movement. When visitors were further away from the enclosure, fewer penguins were huddling, vigilant, and retreating, and more penguins were close to the visitor viewing area, surface swimming and preening in the water. There were no effects of visitor viewing proximity or signage and uniformed personnel on the stress physiology of the little penguins. While there was no effect of signage and uniformed personnel in moderating visitor behaviour, the viewing proximity of visitors to the enclosure affected both visitor and penguin behaviour, where potentially threatening visitor behaviours were decreased and little penguins showed less avoidance behaviour when visitors were further away.

**Abstract:**

We examined the effects of regulating the viewing proximity of visitors and the intensity of visitor behaviour on behaviours indicative of fear and stress physiology of 15 zoo-housed little penguins (*Eudyptula minor*). A 2 × 2 factorial fully randomised design was used to examine the effects of regulating: (1) the viewing proximity of visitors to enclosure, ‘normal viewing distance’ and ’increased viewing distance’ (using a physical barrier set up 2 m away from the enclosure), and (2) the intensity of visitor behaviour, ‘unregulated visitor behaviour’ and ‘regulated visitor behaviour’ (using signage and uniformed personnel). In addition, a treatment consisting of closing the enclosure to visitors was included. Penguin behaviour, visitor numbers and visitor behaviour were recorded by CCTV video recordings and direct observations, respectively. Penguin faecal glucocorticoid metabolites were also analysed as a measure of stress physiology. We found that increased viewing distance reduced (*p* < 0.05) all visitor behaviours except for loud vocalisations and tactile contact with penguins. However, there were no direct effects of signage and uniformed personnel on visitor behaviour (*p* > 0.05). As the regulation of viewing proximity increased from a closed exhibit to an open exhibit with increased viewing distance, and then to an open exhibit with normal viewing distance, this increased the proportion of penguins huddling (*p* = 0.0011), vigilant (*p* = 0.0060) and retreating (*p* = 0.00013), and decreased the proportion of penguins within 1 m of the visitor viewing area (*p* = 0.00066), surface swimming (*p* = 0.00091) and preening in the water (*p* = 0.042). There were also limited effects of regulating visitor behaviour on penguin behaviour. No treatment effects were found on faecal glucocorticoid metabolites (*p* > 0.05). These results indicate that regulating visitor viewing proximity affects penguin behaviours indicative of fear and visitor behaviour. This suggests that close visitor contact can be fear-provoking for little penguins but increasing the distance between visitors and penguins can reduce fear responses of penguins by regulating both viewing proximity and visitor behaviour. However, it is unclear whether these changes in penguin behaviour are due to the increased separation between visitors and penguins and/or specific visitor behaviours associated with close viewing proximity to the enclosure, such as leaning over the enclosure or tactile contact with the pool, which are impeded when visitors are further away.

## 1. Introduction

Understanding human–animal relationships in zoos has been an area of growing importance because of the effects that humans, especially visitors, can have on different zoo species such as non-human primates, big cats and ungulates, varying from negative, positive and neutral effects [1]. However, the manner in which visitors affect zoo animals remains unclear, especially for a rarely studied taxa such as penguins [1,2]. Research on the human–penguin relationship has primarily been conducted on wild penguins, with growing evidence that human contact can negatively affect various wild penguin populations. For example, Adelie penguins (*Pygoscelis adeliae*) have been found to have reduced breeding success as a result of repeated human approaches and nest checks [3]. Yellow-eyed penguins (*Megadyptes antipodes*) at sites exposed to unregulated tourism have been found to have lower breeding success and fledging weights [4,5]. Also, sites frequently visited by tourists dramatically increased heart rate and reduced reproductive success of Humboldt penguins (*Spheniscus humboldti*) [6], and increased vigilance and agonistic behaviours in King (*Aptenodytes patagonicus*), Gentoo (*Pygoscelis papua*) and Royal penguins (*Eudyptes schlegeli*) in response to human approach [7,8]. Furthermore, when exposed to high levels of human visitation, wild little penguin (*Eudyptula minor*) populations in Australia increasingly avoid nesting areas [9]. In contrast, some penguin species in the wild have been found to habituate to human contact, depending on previous experience and the amount of exposure to humans [10,11,12]. Despite the range of research on wild penguins, there is limited research investigating the effects of visitors on zoo-housed penguins.

To our knowledge, there have been only four studies to date that have investigated the effects of visitor presence on zoo-housed penguins. Two of these studies found negative effects of visitors, where one study showed that African penguins (*Spheniscus demersus*) reduced their pond use [13], and another study found that little penguins displayed increased aggression, huddling and avoidance behaviours [14] in response to visitor presence. In contrast, as visitor numbers increased, Humboldt penguins were found to spend more time swimming underwater at the underwater viewing window [15], and aquarium-housed Gentoo penguins increased their behavioural diversity and pool use [16]. These latter two studies suggest a potential positive effect of visitors. However, results from these four studies need to be interpreted with caution as three of the above-mentioned studies examined correlations between visitor numbers and penguin behaviour [13,15,16], and only one study was under experimental conditions [14]. Also, this variation in penguin behaviour related to visitor contact may be due to species or individual animal differences, such as past experiences and coping style, enclosure design or the nature of the visitor interactions [17]. Consequently, it remains unclear what aspect of visitor contact affects penguin behavioural responses.

In zoos, animals may be exposed to a range of stimuli from visitors including auditory, visual, tactile, olfactory and vibratory stimuli that may affect an animal’s perception of visitors, resulting in different behavioural responses [2,18]. If visitors are perceived as fear-provoking, knowledge about the effects of regulating visitor contact, such as visitor behaviour management to reduce stimuli that animals may perceive as threatening, could be useful in reducing visitor effects on animal fear. Behavioural responses to threats, which may be indicative of fear, in penguins commonly include immobility, retreat or avoidance, vigilance, huddling, reduced preening and aggression [7,8,19,20,21,22,23]. Meanwhile, the main physiological stress responses to threatening stimuli are via the sympatho-adrenal medullary system and the hypothalamo–pituitary–adrenal (HPA) axis, resulting in the secretion of catecholamines and glucocorticoids, respectively, when activated [24,25]. Physiological stress in zoo animals is commonly measured through faecal glucocorticoid metabolite (FGM) concentrations because of its non-invasive sampling technique [26]. Consequently, by utilising both animal behavioural and physiological measures, and experimentally regulating visitor contact, we can identify approaches to manage visitors to reduce their behaviours that may be perceived as fear-provoking for penguins, and consequently improve penguin welfare.

In this study, as a follow-up to Sherwen, Magrath, Butler and Hemsworth [14], we jointly examined the regulation of two aspects of visitor contact on little penguin behaviours indicative of fear and stress physiology: (1) visitor viewing proximity, by regulating how close visitors could approach the enclosure, using physical barriers, and (2) intensity of visitor behaviour, by including visual reminders to visitors for them to reduce noise, startling and interactive behaviours. We also added a treatment where the exhibit was closed to visitors.

## 2. Materials and Methods

This study received ethical approval from the Zoos Victoria Animal Ethics Committee (approval number ZV15022) and was conducted at the Melbourne Zoo penguin exhibit. It housed a breeding group of 15 little penguins (ten females, five males; aged between 5 months to 13 years) in an outdoor, naturalistic 330 m^2^ enclosure consisting of sand and vegetation areas, and a large swimming pool (from shallow waters near ramps, to amaximum depth of 3 m) with current flow throughout the water (Figure 1). The exhibit walls were 1.2 m in height and the visitor pathway ran along three sides of the exhibit, with the main penguin viewing positions being along the length of the pool, side A, but opportunities to view penguins also occurred on the short ledge of the pool, side B (Figure 1). The penguins were fed twice a day (09:00 and 15:30). Husbandry including the monitoring of animals, cleaning and feeding of penguins, followed normal routines and remained consistent throughout the course of the study.

### 2.1. Design & Treatments

A 2 × 2 factorial treatment arrangement was used to examine the combined effects of regulating both viewing proximity and the intensity of visitor behaviour on penguins:Viewing proximity of visitors to enclosure at two levels:a.*‘Increased viewing distance’*—a barrier was set up 2 m from the enclosure to increase the distance between visitors and the enclosure (Figure 2). This allowed unrestricted viewing of the enclosure but was a strong impediment to visitors physically interacting with the glass windows of the pool, pool water and other enclosure features.b.*‘Normal viewing distance’*—no barrier was in place and visitors could approach to the edge of the pool (i.e., visitors could approach within 2 m of the enclosure).Intensity of visitor behaviour at two levels:a.*‘Unregulated visitor behaviour’*—visitor behaviours were uncontrolled.b.*‘Regulated visitor behaviour’*—the objective of this treatment was to attempt to reduce the intensity of visitor behaviour using signs requesting visitors to be quiet, move slowly in the exhibit area and avoid physically interacting with the penguins (Figure 3). Also, for this treatment, the researcher was dressed in zoo uniform. This procedure has been successfully used in research on meerkats to reduce the intensity of visitor behaviours [27].

In addition to this factorial arrangement, a treatment consisting of closing the enclosure to visitors (‘closed exhibit’) was included. This fifth treatment can be used to confirm that the presence of visitors increased behaviours indicative of fear in penguins. Alternatively, it can be considered as an extreme level of regulating visitor proximity.

Thus, there were five treatments studied:*Normal viewing distance and unregulated visitor behaviour*—no barrier in place and visitor behaviour was uncontrolled.*Increased viewing distance and unregulated visitor behaviour*—barrier was in place and visitor behaviour was uncontrolled.*Normal viewing distance and regulated visitor behaviour*—no barrier in place and visitor behaviour was attempted to be controlled.*Increased viewing distance and regulated visitor behaviour*—barrier was in place and visitor behaviour was attempted to be controlled.*Closed exhibit*—penguin exhibit closed to the public.

Each treatment was randomly imposed for 2-day periods, two treatments per week with one day break in between (Mon–Tues and Thurs–Fri), with three replicates of each treatment (total of 30 study days) using a fully randomised factorial design. The study was conducted from the end of February to May 2016 (Summer/Autumn) over 9 weeks, and was only conducted on school working days, to avoid the normal systematic variation in visitor numbers that occurs on weekends and during school holiday periods. Two out of the 9 weeks had treatments with no day break in between. This was due to public holidays occurring on the Monday one week and Friday the other week.

### 2.2. Animal Behavioural Observations

Four CCTV cameras placed around the enclosure, covering the main areas utilised by the penguins and recording continuously each day, were used to transcribe penguin behaviour at the group level (Figure 4). Observations were only conducted on the birds visible on camera.

Penguin behavioural data were transcribed during the playback of the video footage and were overlayed to watch all four cameras on one screen using the VLC Media Player 2.2.1 (Figure 4). Observations were conducted on all study days in 3 × 1 h observation blocks (10:00–11:00, 11:30–12:30, 14:00–15:00). A combination of instantaneous point sampling and one-zero sampling was used for behaviour patterns that were considered states (of an appreciable duration) and events (relatively short durations), respectively [28] (see Table 1 for a description of these two types of behaviours). Instantaneous point sampling at 3 min intervals was used to record the number of visible penguins in each behavioural state. Furthermore, one-zero sampling every 30 s, within each 3 min interval, was used to record the number of visible penguins performing each of the behavioural events.

### 2.3. Penguin Faecal Sampling and Analysis: Faecal Glucocorticoid Metabolites (FGM)

Faecal samples were collected by the principal investigator (SJC) at the end of the first and second day of each replicate of each treatment between 15:30 and 16:30, and samples were immediately stored in a freezer at −20 °C. Samples were collected on both days because previous research on this population has found that the median excretion lag time is 7 h and peaked on average at 10.8 h after a health check [29]. However, Sherwen and Fanson [29] found that three of the four penguins studied showed a peak response 5–7 h after the stressor. Consequently, we would expect that if some individuals on the first day of each replicate of a treatment in our experiment were exposed in the morning to fear-provoking bouts of visitor contact, these stress responses would be reflected in FGM concentrations in samples late in the afternoon. Due to limitations of collection, it was not always possible to identify which faecal samples came from which bird. Therefore, we collected all samples deposited over an hour after their feed with the aim of obtaining a good representation of the whole population. An average of 13 samples (range 7–20) were collected per day for the group of 10–15 birds, for an average of 91% of the penguins sampled each day. When more than one sample was knowingly collected from the same individual on the same day (n = 17 samples), the average FGM value for that individual was used. A total of 397 faecal samples were collected during the study. From those samples, 372 samples (96%) were then used to calculate final concentrations for analysis (after samples with a sample coefficient of variation (CV) greater than 20% and sample weights less than 0.01 g were removed); 83, 82, 74, 73 and 60 samples were collected in treatments 1, 2, 3, 4 and 5, respectively.

Faecal samples were dried in a lyophilizer (TFD series, ilShin Biobase, Amsterdam, Netherlands) at −80 °C for 24 h. They were then pounded and sifted to remove debris. One mL of 60% methanol was added to 0.10 g of dried faecal powder in polypropylene tubes. Samples were mixed on a vortex until samples were homogenised (~10 s) and then shaken on an orbital shaker for 1 h then centrifuged for 10 min at 2000 rpm (Hettich Universal 320 R, Tuttlingen, Germany; RCF = 2500). Supernatant was poured into 1.5 mL microcentrifuge tubes and stored in a freezer at −20 °C. FGM was measured and analysed using a validated double-antibody cortisone enzyme-immunoassay for little penguins [29]. Furthermore, this assay was similar to an assay made by the same manufacturer that has been validated for Adelie penguins [30,31]. Assay procedures have been previously described [29]. All samples were assayed in duplicates and data are expressed as ng/g dried faeces for final concentrations. The intra-assay CV was less than 15%. Inter-assay CVs were 4.2% and 9.1% for low and high controls, respectively (n = 11 plates).

### 2.4. Visitor Behavioural Observations

Visitor observations were directly conducted by the principal investigator (SJC) using instantaneous point sampling at 3 min intervals in seven 30 min blocks between 09:30 and 15:15 to record visitor numbers (number of adults and children within 1 m of enclosure). Within each 3 min interval, the frequency of visitor behavioural events (described in Table 2) were recorded continuously in each observation block. These observations were made from the visitor viewing area in the furthest corner of the exhibit area to reduce the effect of researcher presence on the penguins and visitors. Ambient noise level was recorded every 3 min using a sound level data logger (Standard ST-173) which was placed inside the penguin exhibit (Figure 1).

### 2.5. Data Analysis

Measurements were based on the proportion of penguins visible on the video footage and were averaged over the 2-day period treatment to obtain a single summary value for each 2-day period. For animal behavioural states, the proportion of penguins visible per sample point per day was calculated and averaged in order to calculate the proportion of visible penguins observed displaying each behaviour for each day. Animal behavioural events were handled in a similar manner whereby the average proportion of visible birds within each 30 s observation interval per day, observed displaying each behavioural event was calculated. In relation to visitor variables, the numbers of adults and kids were summed to give the visitor number, noise level (dB) was averaged and visitor behaviours were totaled for each day (the average number of times each behaviour was observed in each of the seven 30 min observation blocks) and subsequently averaged over the 2-day period treatment replicates to obtain a single summary value. FGM was also averaged over samples obtained in the 2-day period treatment replicates to obtain a single summary value. The 2-day summary values were used as the unit of analysis in all statistical analysis.

Preliminary analysis indicated that the strongest observed effects could be considered as an ordered function of the allowed viewing proximity of visitors (normal viewing distance vs. increased viewing distance vs. no visitors (exhibit closed)). Thus, measurements were analysed using a single error analysis of variance (ANOVA) with terms for (i) visitor viewing proximity regulation (normal viewing distance; increased viewing distance; exhibit closed), (ii) visitor behaviour regulation (regulated; unregulated) when exhibit was open and (iii) interaction between visitor viewing proximity and visitor behaviour regulation when exhibit was open. These terms had 2, 1 and 1 degrees of freedom, respectively. No visitor measurements were recorded when the exhibit was closed, since most visitor measurements are, by definition, zero (e.g., when the exhibit is closed there are no visitors, thus the frequency of each visitor behaviour is always zero). Thus, the analysis of visitor measurements was modified to have only 2 levels for the visitor proximity term (normal viewing distance; increased viewing distance). In these cases, the analysis simplified to a 2 by 2 (visitor viewing proximity by visitor behaviour regulation) factorial ANOVA. Prior to statistical analysis, the values for animal behaviour (state and events) were angularly transformed and the values for visitor behavioural events were square root transformed so that the residual variation was similar in all treatments. No transformation was required for FGM.

Meteorological data (maximum and minimum daily temperatures), zoo visitor gate numbers and the proportion of male penguins, all measurements that are unlikely to be affected by the treatment applied, were used as covariates if they were significant (*p* < 0.05). This improved, sometimes substantially, the precision of some of the statistical analyses. The proportion of males was considered as a covariate since by the end of the study only females were exhibited after two males were removed for medical reasons and three males were removed to prevent early breeding.

For visitor data, the *p* values, of each term, were calculated using the F distribution with 1 or 8 degrees of freedom, except for analyses that included covariates, where an F distribution with 1 or 7 degrees of freedom was used. Some visitor behaviours only occurred with the normal viewing distance, and in these cases *p* values were calculated using non-parametric permutation tests since the large number of zero values can cause over-sensitivity in the analyses. Also, in these cases, standard errors of difference were calculated from analyses only including normal viewing distances, so as to avoid severe underestimation of these standard errors.

For animal behaviour data and FGM, *p* values for proximity effects were calculated using the F distribution with 2 or 10 degrees of freedom, except for analyses that included covariates, where an F distribution with 1 or 9 degrees of freedom was used. For regulation effects and interactions, *p* values were calculated using the F distribution with 1 or 10 degrees of freedom, except for analyses that included covariates, where an F distribution with 1 or 9 degrees of freedom was used. Analyses were carried out using the ANOVA directive and APERMTEST procedure of GenStat 16th Edition statistical program (software for biosciences).

## 3. Results

### 3.1. Visitor Variables

Increased viewing distance (2 m from the enclosure) reduced visitor tactile contact with enclosure features by 90% (F_1,8_ = 27.71, *p* = 0.00076), looming behaviour with penguins greater than 0.5 m from looming position by 99.8% (F_1,8_ = 762.77, *p* = 3.2 × 10^−9^), tactile contact with water by 100% (F_1,8_ = 104.74, *p* = 0.00054) and sudden movements by 99% (F_1,8_ = 8.67, *p* = 0.019; Table 3). However, there was no corresponding effect (*p* > 0.05) on ambient noise, loud vocalisations, looming behaviour with penguins less than 0.5 m from looming position or tactile contact with penguins. Increased viewing distance was also associated with a 30% reduction in the number of visitors to the exhibit (F_1,7_ = 0.51, *p* = 0.0047; Table 3). There was no evidence at the 5% level that the attempted regulation of visitor behavior affected actual visitor behavior, ambient noise or the number of visitors to the exhibit (Table 4). Nevertheless, the best estimate for the effect of the attempted regulation of visitor behaviour is the frequency of tactile contact with enclosure features (the third highest frequency of the recorded behaviours) which was reduced by about 40%, although the *p* value was 0.30 (Table 3).

### 3.2. Animal Behaviour

#### 3.2.1. Behavioural States

As the viewing proximity of visitors (with unregulated visitor behaviour) increased from a closed exhibit to an open exhibit with increased viewing distance, and then to an open exhibit with normal viewing distance, the proportion of penguins huddling increased from 25% to 69% (F_2,10_ = 14.56, *p* = 0.0011), the proportion vigilant increased from 3% to 22% (F_2,10_ = 8.89, *p* = 0.0060), the proportion of penguins within 1 m of side A (F_2,10_ = 16.63, *p* = 0.00066) and B (F_2,10_ = 11.66, *p* = 0.0024) decreased from 28% to 1% and 3%, respectively, and the proportion surface swimming decreased from 39% to 9% (F_2,10_ = 15.27, *p* = 0.00091; Table 4). When the exhibit was open, regulating the intensity of visitor behaviour (using signage and uniformed personnel) reduced huddling by about 20% (F_1,10_ = 15.95, *p* = 0.0025) but had no effect on vigilance, proximity to side A or B of the visitor viewing area or swimming (Table 4). The proportion of penguins huddling in an open exhibit with regulated behaviour and increased viewing distance (25%) was similar to the proportion huddling when the exhibit was closed to visitors (Table 4).

The effect on locomotion appears more problematic; especially considering the s.e.d.s were considerably lower than those of other behavioural states (Table 4), the significant (*p* < 0.05) effects would disappear if there were similar s.e.d.s to other behavioural states, and the fact that it was the only behaviour in which a significant interaction (*p* < 0.1) between visitor viewing distance and regulated visitor behaviour was found. No other effects of regulating visitor behaviour or interactions were found on penguin behaviour (*p* > 0.05; Table 4). There were no treatment effects (*p* > 0.05) on the proportion of penguins visible, resting, idle and diving (Table 4).

#### 3.2.2. Behavioural Events

Evidence (*p* < 0.05) of visitor viewing proximity on penguin behavioural events only occurred for preening in the water (F_2,10_ = 4.44, *p* = 0.042) and retreating ((F_2,9_ = 28.56, *p* = 0.00013; Table 4). The occurrence of retreating was very low when viewing distance was increased (with unregulated visitor behavior) or when the exhibit was closed (Table 4). On the other hand, preening in the water mostly occurred with visitors at increased viewing distance (with unregulated visitor behaviour; Table 4). Also, there was some evidence (*p* < 0.05) that regulating visitor behaviour decreased the occurrence of allopreening and pecking regardless of viewing distance (Table 4). However, it should be noted that the proportion of penguins engaging in the behavioural events in any 30 s period was, in general, very small.

#### 3.2.3. Little Penguin FGM Concentrations

There was no evidence (*p* > 0.05) of any treatment effect on FGM concentrations (Table 4).

## 4. Discussion

The viewing proximity of visitors was the only factor that had a statistically significant effect on visitor behaviour, whereby the 2 m separation (using a physical barrier) between the visitor viewing area and the penguin enclosure decreased the frequency of tactile contact with enclosure features, looming over the enclosure ledge, tactile contact with the pool’s water and sudden movement. This indicates that the physical barrier not only regulated visitor viewing proximity but also visitor behaviour. Park et al. [32] supports our results, since they found that a physical fence was a more effective strategy to keep visitors on maintained trails at Acadia National Park, USA, compared to educational signage. Similarly, we found no direct evidence that regulating the intensity of visitor behaviour using signage and uniformed personnel was effective, which was aimed at regulating noise, sudden movement and interaction with the penguins. These results suggest that signage and uniformed personnel were much less effective in regulating visitor behaviour than regulating viewing proximity. This is consistent with Acevedo-Gutiérrez et al. [33], who found that posted signs were unsuccessful in managing interactions between tourists and seals. In comparison, other studies have found that signs can effectively regulate visitor behaviour [27,34]. Nevertheless, our findings highlight that physical barriers may be an effective management strategy to manage visitor-penguin interactions in zoos.

The primary objective of our experiment was to identify aspects of visitor contact involved in eliciting behaviours indicative of fear, such as increased avoidance behaviour, huddling and vigilance, in little penguins. The study by Sherwen, Magrath, Butler and Hemsworth [14] found that the presence of visitors was fear-provoking but not specifically what aspect of visitors was responsible for this effect. We found that as the regulation of visitor viewing proximity increased from a closed exhibit to an open exhibit with increased viewing distance, and then to an open exhibit with normal viewing distance, more little penguins were observed huddling and vigilant, and less penguins were observed close to the visitor viewing area and surface swimming. This indicates that the study penguins may have perceived visitors as threatening when in close proximity, with the increased vigilance and huddling behaviour reflecting increased alertness, and the increased distance from the visitor viewing area reflecting avoidance. These results are consistent with Sherwen, Magrath, Butler and Hemsworth [14], who studied the same enclosure but with a different mix of individual little penguins, and studies on wild penguins that have found close human proximity affects their avoidance behaviour. For example, the close proximity of humans to penguin landing sites has been shown to delay the landing times on beaches of Yellow-eyed penguins returning from their daily forage at sea [35], elicit avoidance responses in Adelie [36] and Humboldt penguins [6] and increase heart rates in nesting Gentoo [37] and Adelie penguins [38]. Wild little penguins also prefer to nest further away from visitor paths or areas of high human disturbance [39,40]. Furthermore, in our experiment, the similarity between the effects of increased viewing distance and the exclusion of visitors from the exhibit suggests that a physical barrier to push visitors 2 m away from the enclosure can closely parallel visitor absence for penguins. Consequently, this provides evidence that a physical barrier may be an effective management strategy that results in reduced fear responses in little penguins.

It should be recognised that the increase in behaviours indicative of fear in little penguins, which was elicited by the close proximity of visitors, may have also been influenced by the opportunity for visitors to perform behaviours such as looming, whereby visitors could lean on or over the enclosure ledge, that may have been fear-provoking. This behaviour was the most frequently recorded visitor behaviour and was eliminated by restricting visitors from closely approaching the enclosure by erecting the barrier 2 m from the enclosure. Looming by visitors was often accompanied by tactile contact with the water and sudden movements towards the enclosure. Thus, looming and these accompanying behaviours by visitors when in close proximity are likely to be perceived by penguins as a predatory threat, similar to how an aerial predator would prey upon little penguins. This is supported by Giese [38] that found a human standing lower to nesting Adelie penguins’ horizon represented less of a threat on the basis of reduced heart rate, compared to a human looming over nesting penguins. Furthermore, little penguins are the smallest species of penguins, that grow to an average height of 30 cm and weigh about 1 kg [41,42,43]. They are commonly predated upon by terrestrial predators such as red foxes (*Vulpes vulpes*), cats (*Felis catus*) [44,45] and water rats (*Hydromys chrysogaster*) [46], aerial predators such as sea eagles (*Haliaeetus* spp.) and large gulls [43], and aquatic predators such as long-nosed fur seals (*Arctocephalus forsteri*) [47]. It is therefore not surprising that little penguins may perceive visitors as threatening, particularly when close visitor contact may be intense, unpredictable and possibly unrewarding. In addition to the effects on avoidance behaviour and alertness in little penguins, an interesting finding with the effects of regulating visitor viewing proximity was an increase in the number of birds observed preening in the water when visitors were further away. Sherwen et al. (2015) anecdotally reported that penguins perform increased preening while swimming. Presumably, comfort behaviours such as preening are generally more likely to occur when the animal is not threatened [19,48,49,50]. Furthermore, the deprivation of a presumably highly motivated behaviour such as swimming is a welfare concern [51,52,53]. Wild little penguins spend most of their day(s) foraging out at sea [43,54]. However in the zoo, penguins do not need to forage as their food is provided to them [14]. Yet not surprisingly, we found the penguins still spent most of their time swimming when visitors were not present or when visitors were further away, suggesting it is a highly motivated behaviour and possibly an indicator of a positive welfare state.

While the present experiment and that of Sherwen, Magrath, Butler and Hemsworth [14] show evidence that the penguins perceived visitors as fear-provoking based on behavioural changes, there are several possible explanations for a lack of treatment effects on FGM in our study. There is an extensive body of evidence demonstrating the importance of both predictability and control in determining the behavioural and physiological effects of aversive stimuli [55,56,57]. A degree of both predictability of visitor contact and control over visitor contact, by hiding from and avoiding fear-provoking visitor interactions, may have been an effective adaptive response in ameliorating the activation of the HPA axis in the penguins when visitor viewing proximity and behaviour were uncontrolled (i.e., standard zoo conditions) [58,59]. Another possibility may be that the study period was insufficient to elicit a measurable change in FGM. Our study exposed penguins to some treatments that should have been less stressful than their normal exposure to zoo visitors, and it may take longer for the HPA axis to lower glucocorticoid production than it does to mount a stress response. Nakagawa, Möstl and Waas [30] found that FGM concentrations for Adelie penguins after an adrenocorticotropic hormone (ACTH) challenge can take up to 48 h to return to baseline levels. Consequently, a more sustained period of treatment, that is greater than a 2-day period, may be required to demonstrate increased stress arising from uncontrolled visitor contact.

There were limited effects of attempting to regulate the intensity of visitor behaviour, through use of signage and uniformed personnel, on penguin behaviours indicative of fear. While fewer penguins were observed huddling, allopreening and pecking, there were no effects of attempting to regulate visitor behaviour on the number of penguins observed close to the visitor viewing area, surface swimming and vigilant. These limited effects on penguin behaviours are not surprising, since this treatment had no observed effect on reducing visitor behaviours that intuitively may be fear-provoking such as tactile contact with enclosure features, loud vocalisations, looming over the enclosure ledge, physically touching the pool’s water and suddenly moving towards the enclosure. However, there may be a precision issue with our results regarding the regulation of visitor behaviour, as the frequency of tactile contact with enclosure features, loud vocalisations and physical contact with the pool were about halved when signage and uniformed personnel was used. Despite not being statistically significant, it could explain why there were still some effects of attempting to regulate visitor behaviour on some penguin behaviours which provides evidence that signage and uniformed personnel must have affected some relevant aspect of visitor behaviour. On the other hand, a possible reason that the signage had limited effectiveness is that some of the potentially fear-provoking visitor behaviours were not specifically targeted in the signage. Several studies have shown signs that have clear instructions that target a specific behaviour [60], justification [61] and appeal to visitors’ emotions [34], are more effective in moderating visitor behaviour. The signs, in our study, requested visitors to be quiet, move slowly and not interact with the penguins in the exhibit area, but there was no mention of avoiding specific behaviours such as looming and tactile contact with the water, which were effectively reduced or eliminated by regulating viewing proximity. Thus, in addition to close proximity per se, these visitor behaviours that are possible when close to the enclosure such as leaning on or over the enclosure ledge and physical contact with the water, may be fear-provoking and may be more effectively managed by using a physical barrier rather than signage. Clearly, further experimentation is required to fully appreciate visitor behaviours that are fear-provoking for penguins.

The decreased number of visitors found when visitor viewing distance was increased suggests a potential reduction in visitor dwell times and possibly reduced visitor viewing experience at the penguin enclosure. This is consistent with Sherwen, Magrath, Butler, Fanson and Hemsworth [18] who found that reducing the visual contact of capuchins with visitors, by using a one-way visual barrier, reduced visitor numbers. In contrast, Sherwen, Magrath, Butler, Phillips and Hemsworth [27] found no effect of signs on visitor numbers. Further investigation on the effects of manipulating visitor behaviour on visitor experience would be valuable for future focus.

Interestingly, no treatment effects were found on the proportion of penguins visible despite the evidence that visitors may be perceived as threatening for penguins with close visitor contact. Visitor gate numbers was a statistically significant covariate for this behaviour. The more visitors that came through the zoo gates, the less penguins were visible. This suggests that penguins may utilise different behavioural strategies depending on the type of visitor contact. For example, high numbers of visitors in the zoo would increase noise levels, but not direct interactive behaviours that would occur at the exhibit. It would be valuable to further investigate this relationship.

## 5. Conclusions

This study provides evidence that regulating the viewing proximity of visitors, either by the installation of a physical barrier or by closing the exhibit to visitors, strongly reduces little penguin behaviours indicative of fear and strongly increases swimming, which is likely impeded by fear. Regulating visitor behaviour by signage was much less effective in changing penguin behaviour, but it is unknown to what extent this was caused by having signage that insufficiently targeted specific visitor behaviours. Regulating visitor viewing proximity or visitor behaviour did not affect little penguin FGM, which may suggest that penguin behavioural responses such as avoidance may have been an effective adaptive response in ameliorating stress arising from close visitor contact, or that the treatment period was insufficient to elicit a measurable change in FGM. However, whether it is visitors being further away per se and/or the reduction of some visitor behaviours such as leaning over the enclosure or tactile contact with the pool that are impeded when visitors are further away, remains unclear.

## Figures and Tables

**Figure 1 animals-09-00285-f001:**
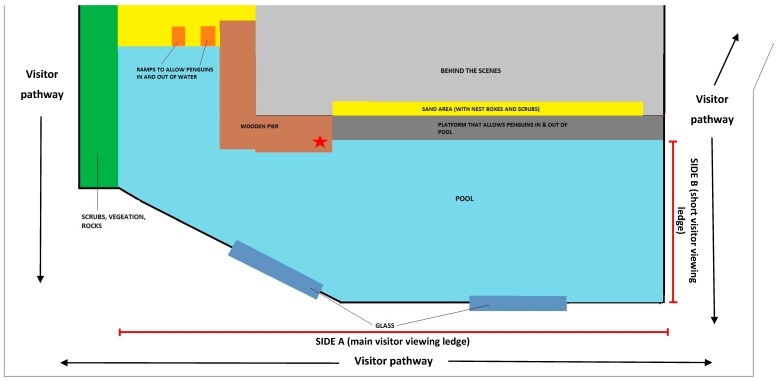
Diagram of Melbourne Zoo penguin exhibit. Red star indicates the position of sound level meter.

**Figure 2 animals-09-00285-f002:**
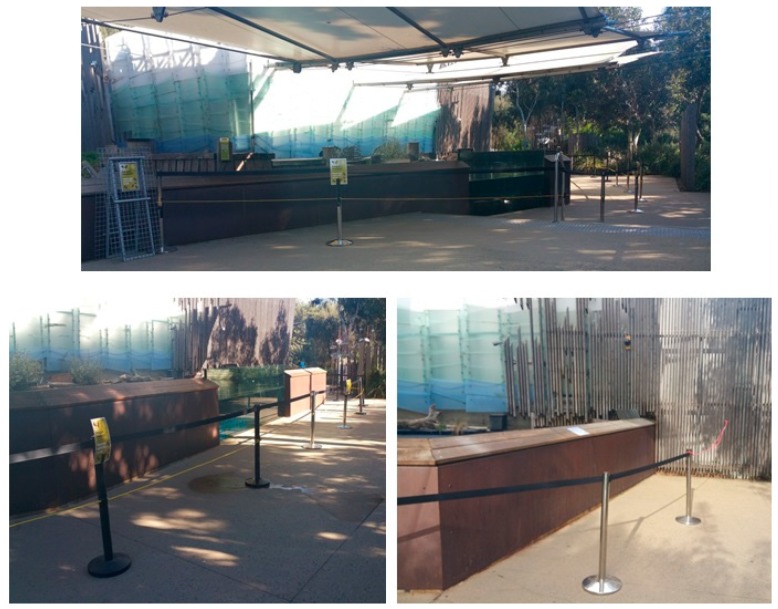
Setup of the barrier in the exhibit area.

**Figure 3 animals-09-00285-f003:**
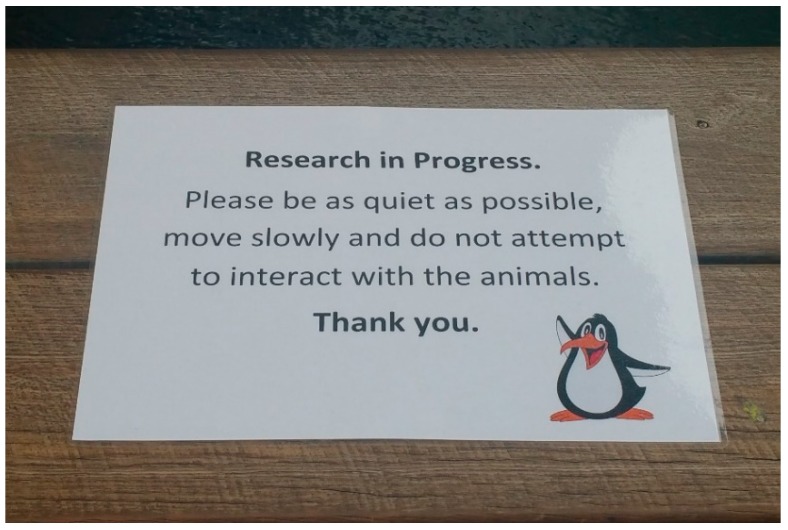
Signs used to attempt to regulate visitor behaviour.

**Figure 4 animals-09-00285-f004:**
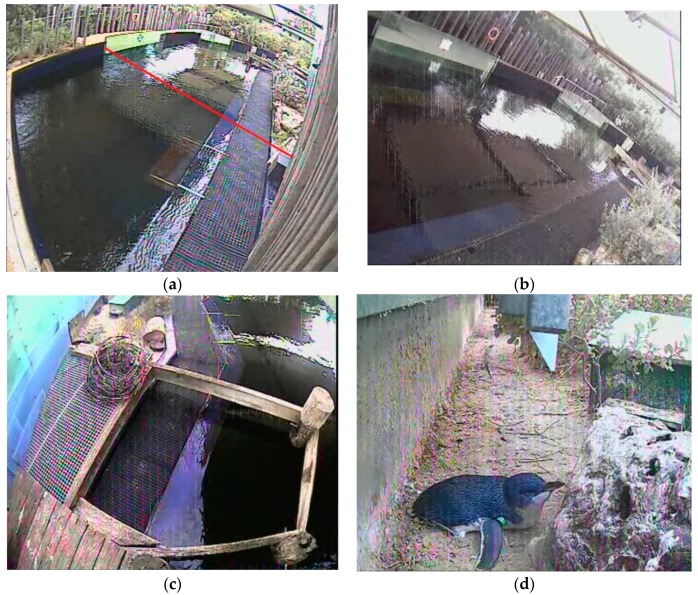
The areas studied, captured by the cameras: (**a**) Area 1—captured the main pool corner of the exhibit where penguins were clearly visible to visitors in this area (the red line indicates where Area 1 ended for observations due to overlap between the cameras for Area 1 and 2) (**b**) Area 2—captured the main length of the pool (penguins in this area were clearly visible to visitors), (**c**) Area 3—captured the pier and a section of the pool (penguins were clearly visible to visitors in this area) and (**d**) Area 4—captured the land area of the exhibit (penguins in this area were either not visible to visitors or partially obstructed by exhibit features including vegetation, wooden barriers, nest boxes).

**Table 1 animals-09-00285-t001:** Ethogram of penguin behaviour.

Behaviour	Description
*States*	
Distance from visitor viewing area (m)	Side A: long/main visitor viewing ledge of the exhibit, within <1 m or >1 m of visitor viewing area. Side B: short visitor viewing ledge of the exhibit, within <1 m or >1 m of visitor viewing area.
Huddling (land)	Stationary, positioned within one flipper distance of at least one other penguin.
Resting (land)	Belly on the ground, in a prone position, with eyes open or closed.
Idle (land)	Standing on two feet with a relaxed posture, eyes open or closed; not visually scanning the environment.
Vigilance (land)	Standing on two feet, visually scanning the environment with head movements from left to right or vice versa.
Locomotion (land)	Upright position, moving from one location to another either by walking or running.
Surface swimming	Moving or floating on the surface of the water, with head erect or in the water.
Diving	Swimming under the water surface.
*Events*	
Preening (on land)	Running bill through plumage on land.
Preening (in water)	On the surface of the water and running bill through plumage.
Allopreening	Running bill through the plumage of another bird(s).
Agonistic interactions (one or a combination of these behaviours)	Peck: Directed at another individual in which an individual directly hits or strikes at another bird with its bill. Bill slap: hit or strike with the side of the bill.Bill joust: an individual interlocks bill with another individual’s bill.Lunge: sudden forward thrust of the body towards another individual.Chase: individual runs after another individual.
Flee	Diving or moving rapidly away from current position in response to a direct approach or interaction by human.
Retreat	Swimming or locomoting (on land) slowly away from human approach or presence.
Social interaction	On land: Approaching another individual with flippers back, pushing individual in a circular motion, nibbling and/or preening head or neck of the individual. In water: approaching another individual in the water, in a circular motion, nibbling and/or preening head or neck of the individual.
Manipulating inanimate object	Using bill to peck or nibble at an inanimate object such as a plant, stick, rock, grass, etc.
Chasing insect	Following a flying insect.
Interaction with keeper/staff	Approaching keepers and engaging in play, feeding and/or agonistic behaviours toward keepers.

**Table 2 animals-09-00285-t002:** Ethogram of visitor behaviours.

Behaviour	Description
Tactile contact with enclosure features	Tapping or banging on the glass windows, barriers, or enclosure features.
Loud vocalisations	Shouts, screams, loud whistles to attract the animals’ attention.
Looming	Leaning on or over the exhibit barriers to view animals in the water or on land. Penguins > 0.5 m—penguins greater than 0.5 m away from looming position. Penguins < 0.5 m—penguins less than 0.5 m away from looming position.
Tactile contact with water	Touching/flicking/slapping the water with the hand(s), which creates ripples in the water.
Tactile contact with penguin	Touching penguin with the hand(s).
Sudden movement	Running, waving or jumping towards or at the penguin(s) and/or exhibit.

**Table 3 animals-09-00285-t003:** Effect of treatment on visitor variables measured. The means shown are the mean frequency of visitor behaviours (per seven 30 min observations blocks per 2-day period) considered as events and ambient noise level per 2-day period. All measurements were square root transformed, except for ambient noise level and the number of visitors (*). Back-transformed means are presented in parentheses.

Visitor Variables	Covariate	Open Exhibit					*p*-Value		
		Increased Viewing Distance		Normal Viewing Distance			Proximity ^a^	Regulation Effects ^a^	Proximity × Regulation Interaction
		Regulated	Unregulated	Regulated	Unregulated	s.e.d. ^c^			
Ambient noise (dB) *	Maximum temperature	62	61	61	62	0.62	0.49	1.00	0.11
Number of visitors *	Visitor gate numbers	360	300	500	460	46	**0.0047**	0.080	0.77
***Behaviour***									
Tactile contact with enclosure features		0.91 (0.83)	1.7 (2.8)	5.1 (26)	6.2 (39)	1.18	**0.00076**	0.30	0.84
Loud vocalisations	Proportion of males	8.5 (73)	9.8 (97)	7.0 (48)	11 (123)	1.87	0.98	0.075	0.33
Looming with penguin > 0.5 m (from looming position)		0.57 (0.32)	1.6 (2.6)	22 (492)	21 (461)	1.06	**3.2 × 10^−9^**	0.82	0.28
Looming with penguin < 0.5 m (from looming position)		0	0	0.86 (0.74)	1.3 (1.6)	0.85 ^d^	0.061 ^b^	0.64 ^b^	0.64 ^b^
Tactile contact with water		0	0	3.6 (13)	5.1 (26)	0.85 ^d^	**0.00054** ^b^	0.09 ^b^	0.09 ^b^
Tactile contact with penguin		0	0	0	0.64 (0.41)	0.36 ^d^	0.18 ^b^	0.18 ^b^	0.18 ^b^
Sudden movement		2.0 (4)	1.1 (1)	5.0 (25)	5.0 (25)	1.63	**0.019**	0.72	0.73

^a^*p* values were calculated using F tests based on 1, 7 or 1, 8 degrees of freedom depending on whether a covariate was included in the analysis. *p* values less than 0.05 are in bold. ^b^
*p* values calculated using permutation test. ^c^ s.e.d. denotes standard error of difference. ^d^ sed calculated using residual standard deviation obtained from analysis only using observations with normal viewing distance.

**Table 4 animals-09-00285-t004:** Effect of treatment on the mean proportion of visible penguins (%) performing each behaviour (state and event) per 2-day period and stress physiology. Results are the average proportion of visible penguins performing each behavior. Data were angularly transformed, except for FGM concentrations. Back-transformed means are presented in parentheses.

Behaviour	Covariate	Closed Exhibit	Increased Viewing Distance		Normal Viewing Distance		s.e.d. ^c^	*p* Value		
								Proximity ^a^	Regulation when Exhibit Open ^b^	Proximity × Regulation Interaction
			Regulated	Unregulated	Regulated	Unregulated				
*States*										
Penguins visible	Visitor gate numbers	40 (41)	37 (36)	43 (46)	36 (35)	39 (40)	3.3	0.77	0.070	0.48
Huddling	-	30 (25)	30 (25)	40 (42)	41 (43)	56 (69)	4.5	**0.0011**	**0.0025**	0.50
<1 m from side A	-	32 (28)	22 (14)	22 (14)	7 (2)	7 (1)	5.4	**0.00066**	0.93	0.95
<1 m from side B	-	32 (28)	27 (21)	28 (21)	8 (2)	11 (4)	6.4	**0.0024**	0.69	0.75
Resting	-	18 (10)	24 (17)	19 (11)	19 (10)	24 (16)	4.5	0.70	0.96	0.15
Idle	Maximum temperature	36 (35)	34 (31)	38 (38)	40 (41)	45 (51)	4.2	0.081	0.14	0.91
Locomotion (on land)	Proportion male	15 (7)	15 (7)	15 (7)	19 (11)	12 (5)	1.7	0.85	**0.023**	**0.019**
Vigilant	-	10 (3)	19 (10)	19 (11)	25 (17)	28 (22)	4.5	**0.0060**	0.53	0.60
Surface swimming	-	38 (39)	33 (30)	31 (26)	16 (7)	17 (9)	5.1	**0.00091**	0.92	0.61
Diving	Proportion male	13 (5)	12 (4)	13 (5)	8 (2)	9 (2)	2.6	0.077	0.69	0.83
*Events*										
Preen (land)	-	19 (10)	18 (9)	18 (9)	24 (16)	22 (15)	4.2	0.21	0.84	0.83
Preen (water)	-	9 (3)	8 (2)	11 (4)	6 (1)	4 (0.54)	2.4	**0.042**	0.75	0.27
Allopreen	-	6 (1)	4 (0.54)	8 (2)	6 (1)	8 (2)	1.7	0.57	**0.039**	0.45
Peck	-	3 (0.31)	2 (0.18)	3 (0.36)	3 (0.33)	5 (0.62)	0.54	0.074	**0.014**	0.78
Agonistic interactions	-	4 (0.48)	4 (0.39)	4 (0.58)	4 (0.51)	5 (0.74)	0.63	0.47	0.10	0.92
Flee	Proportion of males	2 (0.093)	0.54 (0)	0.13 (0)	2 (0.098)	1 (0.061)	0.77	0.069	0.48	0.98
Retreat	Proportion of males	0.36 (0)	0.17 (0)	0.53 (0)	2 (0.16)	2 (0.14)	0.38	**0.00013**	0.65	0.39
Social interaction (land)	Proportion of males	3 (0.24)	3 (0.28)	4 (0.37)	2 (0.17)	1 (0.020)	1.1	0.11	0.52	0.23
Social interaction (water)	-	0.73 (0.016)	1 (0.052)	1 (0.035)	0	0.32 (0)	0.96	0.35	0.95	0.69
Manipulate object	-	6 (1)	6 (1)	5 (0.76)	5 (0.74)	3 (0.35)	1.8	0.52	0.41	0.74
Chase Insect	-	1 (0.059)	0.58 (0.010)	1 (0.037)	0.36 (0)	0.32 (0)	0.73	0.29	0.65	0.60
Interaction with keeper	Maximum temperature	0	0.59 (0.011)	0.10 (0)	0.28 (0)	0.67 (0.014)	0.45	0.43	0.67	0.20
*Physiology*										
FGM concentration (ng/g)	Maximum temperature	1600	1140	1657	1272	1600	284	0.71	0.060	0.67

^a^ Closed vs Increased viewing distance vs Normal viewing distance. *p* values are calculated using F tests based on 2, 9 or 2, 10 degrees of freedom depending on whether a covariate is included in the analysis. *p* values less than 0.05 are in bold. ^b^ Calculated F tests for Open exhibit based on 1, 9 or 1, 10 degrees of freedom depending on whether a covariate is included in the analysis. ^c^ s.e.d. denotes standard error of difference.

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
