# Peer review of "Effects of Regulating Visitor Viewing Proximity and the Intensity of Visitor Behaviour on Little Penguin (Eudyptula minor) Behaviour and Welfare"

_animals, 2019, doi:10.3390/ani9060285_

Round 1
Reviewer 1 Report
I consider this a very interesting paper with a well-done research about the effect of visitors on little penguin behaviour and welfare. I just have a few comments/suggestions:
Line 148: seeing the results and that the visitor behaviours were not really reduced I think you need to rephrase ‘the intensity of visitor behaviours at the viewing area was reduced…’. In lines 166 and 168 it is mentioned that ‘visitor behaviours were attempted to be controlled’. This sounds more in line with the reality of the situation.
L 149-151: How active was the control of visitor behaviour by the researcher? Was it just the signs that requested visitors to be quiet and move slowly or also the researcher was giving this guidance to the visitors? The way it is written now I understand that both the researcher and the signs were requesting visitors to be quiet, move slowly and not attempt to interact with the penguins.
L 185-186: Does this mean that observations were not done at individual level? This sentence could be simplified while keep the comment that observations were only conducted on birds visible on camera.
L201: Change ‘was’ for ‘were’.
L 209-211: What was decided to collect samples on the first day of each replicate if the lag time and peak after the stressor did not happened yet when these samples were collected?
L 229: The reference of ‘Sherwen and Fanson, 2015’ should be added as the rest of references in the text: [number].
L 245 (Table 2): Add a period after the description of the behaviour ‘Loud vocalisations’.
L 415: Only the first word of the scientific name of the red fox should be capitalised: Vulpes vulpes.
L 416: If more than one species of sea eagles is mentioned in the text next to the scientific name it should be added spp.: Haliaeetus spp.
L 497: On the conclusions I would add a short sentence where you mention the results obtained on the FGM.
L 524 (References): All references should be re-checked for format and typos. For example: Reference number 7: Polar Biology (instead of BIology).
At least references number 10, 11, 12, 13, 14, 17, 29, 30, 31, 36, 38, 40, 41, 42, 44, 47 and 55 have mistakes on the way the scientific name should be written down: Genus species.
At least references number 15, 39, 40, 43, 44, 45, 47 and 61 should have toponyms written down with capital letter, i.e. Iceland instead of iceland.
All references should be re-checked.
Author Response
Responses to Reviewer 1 comments have been attached.

Reviewer 2 Report
Thank you for the opportunity to review this manuscript. In general, I think that the authors have done a nice job of executing a study investigating the influence on visitor proximity and behavior on the behavior of little penguins. The behavior data collection and analysis methods are appropriate, and the conclusions drawn with respect to behavioral outcomes are well-reasoned.
The main issues with this paper is the inclusion of FGM analysis and they are significant enough that I would only recommend publication if the entire FGM section is removed. One of the main reasons that the FGM portion of the study is unsuitable for publication is actually acknowledged by the authors in the discussion. The "control" state for these penguins is the condition where visitors are always able to be within close proximity and perform the types of behaviors found to evoke behaviors indicative of fear and the "treatments" were conditions thought to decrease visitor access. However, a 2-day study period is, as acknowledged, unlikely to be long enough to allow for readjustment of the HPA activity especially when we don't know how the other physical/ environmental factors that were present during the study period (for example social changes/poor health of some males was mentioned) could have influenced FGMs. Also, since the penguins are chronically exposed to conditions where that elicit fearful behaviors it is likely that they have experienced chronic activation of the HPA axis. Such chronic activation can be associated with glucocorticoid levels returning to baseline levels even as the stressors persist in the environment and when behaviors still are indicative of "stress". This resetting of the HPA axis (or allostasis)can give the appearance of "normal" cortisol levels and can be very difficult to detect without specific protocols.
HPA axis activity is very sensitive to factors such as temperature, feeding, health status and is well-known to show individual differences related to age, sex, and experience. Pooling of data greatly reduces the ability to detect meaningful changes in FGMs because it greatly reduces the ability to consider these environmental/individual factors in interpretation. I appreciate the logistical challenges of collecting individually identifiable samples, but the extra effort to do so is essential if we hope to use FGM as a tool in understanding individual animal welfare.
The methods section states that the fecal samples were collected "at the end of the day". It is unclear for what duration (or range of durations) the samples remained at "room temperature" before being frozen, but best practice is to freeze samples immediately because changes in concentration of metabolites can occur rapidly.
Finally, the "coping" explanation given for a lack of difference in FGM between conditions is irresponsible given the weaknesses in the FGM methodology. This explanation could be interpreted to mean that the behaviors exhibited by the penguins did not have meaning in the context of animal welfare because they served to mitigate changes in stress physiology. In drawing this conclusion, you've undermined the interesting and important results of the behavioral study which clearly indicates that visitor proximity/behavior causes behavioral changes indicative of fear/distress and demonstrates that changes should be made to how guests access this exhibit. Along those lines, please add an update to the conclusions indicating what, if any, permanent changes have been made as a result of this study. It is important that papers presenting zoo-based research discuss actual changes to management and husbandry when evidence suggests welfare could be improved by such changes...or else, what is the point of the study?
Author Response
Responses to Reviewer 2's comments have been attached/uploaded as a pdf.

Reviewer 3 Report
The paper critically investigated the intensity of visitors' influence on little penguins behaviour and welfare in a zoo setting. The combined effects of regulating both viewing proximity and visitors behaviours were studied with a 2x2 factorial treatment arrangement. The study correctly used both behavioral observations and physiological measures to assess animal welfare. The study is well designed and results clairly explained. The conclusions highlighted that visitors strongly affect the behaviour of the species, even if physiological measures did not support these findings. This is consistent with past studies on penguins that reported a decreased functional capability of the adrenocortical tissue after few days of stress (Walker & al., 2006). Considering that the more stressful situation is the usual one for the penguins, it is not surprisingly that authors did not find any significant difference due to very short periods of an ameliorating condition.
The study is a follow-up of a previous study made in the same zoo with the same species (Sherwen, Magrath, Butler and Hemsworth, 2015) where significant negative outcomes on penguins had been highlighted because of visitors' influence. So, apart from its scientific value, some important ethical concerns that resulted from the study have to be highlighted and some revisions on the paper in consideration of them are suggested:
1) Being a follow-up of a previous study, changes that have been eventually implemented to improve the welfare standards of the animals should be reported in this study.
2) Considering that the same welfare issues had been identified three years before, authors should discuss why they did not include a third phase in the experiment in which practical measures were implemented in order to face the problem.
3) In the discussion session, reasons for the continuing of the problem after the first paper and possible solutions should at least be discussed in depth. Scientific research made in zoos setting should at least be useful to tackle welfare problems if detected.
Walker, B. G., Dee Boersma, P., & Wingfield, J. C. (2006). Habituation of adult Magellanic penguins to human visitation as expressed through behavior and corticosterone secretion. Conservation Biology, 20(1), 146-154.
Author Response
Responses to Reviewer 3's comments have been attached/uploaded as a pdf.
